# Placenta Transcriptome Profiling in Intrauterine Growth Restriction (IUGR)

**DOI:** 10.3390/ijms20061510

**Published:** 2019-03-26

**Authors:** Marta Majewska, Aleksandra Lipka, Lukasz Paukszto, Jan Pawel Jastrzebski, Karol Szeszko, Marek Gowkielewicz, Ewa Lepiarczyk, Marcin Jozwik, Mariusz Krzysztof Majewski

**Affiliations:** 1 Department of Human Physiology, School of Medicine, Collegium Medicum, University of Warmia and Mazury in Olsztyn, Warszawska Str 30, 10-082 Olsztyn, Poland; ewa.lepiarczyk@uwm.edu.pl (E.L.); mariusz.majewski@uwm.edu.pl (M.K.M.); 2 Department of Gynecology and Obstetrics, School of Medicine, Collegium Medicum, University of Warmia and Mazury in Olsztyn, Niepodleglosci Str 44, 10-045 Olsztyn, Poland; aleksandra.lipka@uwm.edu.pl (A.L.); marekgowkielewicz@gmail.com (M.G.); prof.jozwik@gmail.com (M.J.); 3 Department of Plant Physiology, Genetics and Biotechnology, Faculty of Biology and Biotechnology, University of Warmia and Mazury in Olsztyn, Oczapowskiego Str 1A, 10-719 Olsztyn-Kortowo, Poland; pauk24@gmail.com (L.P.); bioinformatyka@gmail.com (J.P.J.); 4 Department of Animal Anatomy and Physiology, Faculty of Biology and Biotechnology, University of Warmia and Mazury in Olsztyn, Oczapowskiego Str 1A, 10-719 Olsztyn-Kortowo, Poland; karol.szeszko@uwm.edu.pl

**Keywords:** IUGR, placenta, RNA-seq, SNV, alternative splicing, transcriptome

## Abstract

Intrauterine growth restriction (IUGR) is a serious pathological complication associated with compromised fetal development during pregnancy. The aim of the study was to broaden knowledge about the transcriptomic complexity of the human placenta by identifying genes potentially involved in IUGR pathophysiology. RNA-Seq data were used to profile protein-coding genes, detect alternative splicing events (AS), single nucleotide variant (SNV) calling, and RNA editing sites prediction in IUGR-affected placental transcriptome. The applied methodology enabled detection of 37,501 transcriptionally active regions and the selection of 28 differentially-expressed genes (DEGs), among them 10 were upregulated and 18 downregulated in IUGR-affected placentas. Functional enrichment annotation indicated that most of the DEGs were implicated in the processes of inflammation and immune disorders related to IUGR and preeclampsia. Additionally, we revealed that some genes (*S100A13*, *GPR126*, *CTRP1*, and *TFPI*) involved in the alternation of splicing events were mainly implicated in angiogenic-related processes. Significant SNVs were overlapped with 6533 transcripts and assigned to 2386 coding sequence (CDS), 1528 introns, 345 5’ untranslated region (UTR), 1260 3’UTR, 918 non-coding RNA (ncRNA), and 10 intergenic regions. Within CDS regions, 543 missense substitutions with functional effects were recognized. Two known mutations (rs4575, synonymous; rs3817, on the downstream region) were detected within the range of AS and DEG candidates: *PA28β* and *PINLYP*, respectively. Novel genes that are dysregulated in IUGR were detected in the current research. Investigating genes underlying the IUGR is crucial for identification of mechanisms regulating placental development during a complicated pregnancy.

## 1. Introduction

Intrauterine growth restriction (IUGR) is a serious pathological complication associated with compromised fetal development during pregnancy [1,2]. By definition, the ultrasound estimated fetal weight (EFW) is below the 10th percentile, in relation to the gender, race and genetic growth potential, as per gestational age [3,4,5]. Generally, IUGR may evolve in asymmetrical (70–80% of cases) or symmetrical (20–30% of cases) manner [6], and may be diagnosed as early or late-onset [7]. To examine the fetal growth, serial of typical ultrasonographic biometric measures as biparietal diameter, head, and abdominal circumference, femur length, jointly with monitoring of fetal anatomy are commonly applied [8]. Additional evaluations such as amniotic fluid assessment, fetal vessels (especially: umbilical artery, medial cerebral artery and ductus venosus) and maternal (uterine arteries) Doppler velocimetry are applied in-depth analysis as an indicator of IUGR [9,10]. Clinically, IUGR refers either to prenatal condition or neonates born with sights of malnutrition and in utero growth retardation [11]. However, the universally accepted definition and a consensus diagnosis of IUGR do not exist [10,12,13].

Adequate fetal growth and proper functioning of the placenta is a valuable predictor of pregnancy outcome [1]. Hence, impaired fetal growth is one of the most important causes of prematurity, stillbirth and infant mortality [14,15]. Moreover, growth-restricted newborns are susceptible to pulmonary hypertension, hypothermia, hypo-, hyper-glycemia, pulmonary hemorrhage [16], and may suffer from cognitive delay, neurological, and psychiatric disorders in childhood [17,18,19]. Adults are prone to obesity, hypertension, type-2 diabetes, as well as neurological, cardiovascular, renal, hepatic, and respiratory diseases [20,21,22,23]. Etiology of IUGR is multifactorial and comprises a wide range of various either maternal, fetal, placental or genetic causes [4]. Maternal risk factors include parasite infections (TORCH, malaria), poor maternal health (malnutrition, diabetes mellitus, hypertension, anemia, cardiac, hepatic and chronic renal diseases), obesity (BMI > 30), drug abuse, smoking, female age above 35 years, and multiple gestations [6,24]. Fetal factors encompass chromosomal abnormalities, as at least 50% of fetuses with trisomy 21, 13, or 18 or Turner’s syndrome are associated with higher rate of fetal growth restriction (FGR) [25]. Moreover, specific placental disorders like infarction, fetal vessel thrombosis, preeclampsia (PE), decidual or spiral artery arteritis, chronic villitis, placental hemangioma, as well as umbilical cord abnormalities like velamentous or marginal cord insertion also affect IUGR initiation [6,26]. FGR and related placental pathologies occur when inadequate remodeling of maternal spiral arteries leads to a high-resistance low-flow system of uteroplacental circulation [27]. Nevertheless, among all causes, uteroplacental insufficiency, usually related to incorrect implantation of the placenta, is assumed to be the most common cause of IUGR [28]. The placenta is essential for the development of the fetus and successful pregnancy outcome serves as an organ for nourishing, protection, and gas exchange, constantly supports and enhances the growth of the fetus [29]. Placenta produces and secretes both growth factors and hormones that regulate fetal growth [30]. The uteroplacental insufficiency that interrupts oxygen and nutrients supply to the fetus results in abnormal fetal growth and creates conditions for IUGR initiation [24]. However, sometimes IUGR develops in healthy, young, slim pregnant women who do not manifest any known risk factors or addictions that may be connected with incorrect implantation, which relies on genes expression. 

Development and function of the placenta are under influence of sophisticated pathways related to the expression of substantial genes throughout pregnancy progress [31,32]. The placenta has a unique transcriptional landscape [33,34], therefore, any alterations in gene expression or posttranslational modification may disturb various critical functions, manifested in pathologies and disorders during pregnancy [35]. Consequently, impaired expression of fetal, maternal or placental genes affects fetal growth deceleration that underlies course of IUGR [6,36]. Mutations, polymorphisms, alternative splicing (AS) events, disturbed over or under-expression of specific placental genes - all of them may also be responsible for uteroplacental insufficiency leading to abnormal fetal growth and development [4]. The previous analysis enabled insight into the general human placental transcriptome of the term placenta [33,34]. The goal of the current study was to broaden knowledge about the transcriptomic complexity of the human placenta by identifying genes potentially involved in IUGR pathophysiology.

## 2. Results

### 2.1. Statistics of the Placenta Transcriptome 

The RNA-seq of the constructed cDNA libraries allowed the characterization of the transcriptome profile of IUGR-affected human placenta (Figure 1).

High-throughput sequencing on an Illumina platform generated 546,017,586 raw paired-end reads. After trimming, 477,019,796 reads with good quality were obtained, of them, 419,725,222 were uniquely mapped. Moreover, 8.8% of reads were mapped to multiple loci (Table 1). For all input data, the mean percentage distribution of aligned bases was as follow: 2% were derived from intergenic, 8% from intronic, 35% from untranslated regions, and 55% from coding regions (Appendix A). The total number of 37,501 transcriptionally active regions, including protein-coding genes, were expressed in at least half of the placental libraries. 

### 2.2. Differentially-expressed Genes (DEGs) and Functional Annotations

Two algorithms, DESeq2, and edgeR revealed 39 and 60 differentially-expressed genes (adjusted *p*-value < 0.05), respectively. Normalized counts and logarithmic values of fold change (log2FC) were visualized in MA (Figure 2A) and Volcano plots (Figure 2B). 

Merging of these methods allowed the selection of 28 differentially-expressed genes (DEGs), between IUGR-affected and control libraries (Figure 3A; Table 2). Among the identified DEGs, 10 were upregulated and 18 were downregulated in IUGR-affected placentas. 

Expression profiles of DEGs were prepared for all biological replicates and presented on heatmap (Figure 3B). The five upregulated protein-coding genes were: *ARMS2*, *ASTE1*, *ADAM2*, *TCHHL1*, *BTNL9*. Five the most downregulated gene were: *THEMIS*, *PTPRN*, *FNDC4*, *SIRPG*, *SLC38A5* (Table 2).

The 21 out of 28 DEGs were assigned to 10 main Gene Ontology (GO) categories. Only underexpressed DEGs, engaged mainly in T cell regulation and in the immune response, created biological networks that were qualified as significant. (Table 3). Functional enrichment annotation indicated, that most of the DEGs were implicated in the processes of inflammation and immune disorders (*IL7R*, *PINLYP*, *FNDC4*, *ARMS2*, *LCK*, *ZAP70*, *BCL11B*, *SIRPG*, *ITK*, *BTNL9*) related with IUGR and preeclampsia (PE). Enrichment Kyoto Encyclopedia of Genes and Genomes (KEGG) analysis revealed DEGs engaged into three metabolic pathways: NF-kappa B signaling pathway (*ZAP70*, *LCK*, *LTB*), T cell receptor signaling pathway (*ITK*, *ZAP70*, *LCK*), and primary immunodeficiency (*ZAP70*, *IL7R*, *LCK*).

### 2.3. Alternative Splicing Patterns

Comparing IUGR-affected and control samples the short reads were used to measure percent splicing inclusion (PSI) of the alternative exon, or changes of its inclusion/exclusion (ΔPSI). Alternative splicing events (ASEs) were investigated by rMATs software that permitted to identify 37,527/38,956 splice junctions and reads on target (ROT + JC) and splice junctions only (JC) ASEs and within this group, 2,997/3,691 (ROT+JC/JC) were qualified as differential alternative splicing (DAS) events (Appendix A, Figure 4A). Simultaneously, SUPPA method indicated 90,233 ASEs, among them 130 events were significant DAS (Appendix A, Figure 4B,C). Finally, 11 DAS genes were detected by both methods and intersected according to genomic localization. Among them 26 DAS events were assigned to 20 skipping exon (SE), three (alternative 3′splice site) A3SS, and three (retention intron) RI by rMATS (Appendix A). 

In the same pool of genes, SUPPA described 20 DAS including 10 alternative first exon (AF), 8 SE, one alternative last exon (AL) and mutually exclusive exons (MXE) events (Table 4). In the case when both methods predicted the different type of events, we decided to consider SUPPA results as a more precise approach (recover additional types of AS events—AF and AL). Our findings revealed that some genes (*S100A13*, *GPR126*, *CTRP1*, and *TFPI*) involved in the alternation of splicing events were mainly implicated in angiogenic-related processes. The four alternative exons (skipping events) within *SCEL* (ΔPSI = 0.36), *SIN3B* (ΔPSI = 0.09), *CTRP1* (ΔPSI = 0.24), and *TFPI* (ΔPSI = 0.13) tended to be highly included in IUGR samples (Table 4). Moreover, higher percentage of exclusion were calculated within *PA28β* (ΔPSI = −0.29), *PSAP* (ΔPSI = −0.1), *EVI5* (ΔPSI = −0.31), and *GPR126* (ΔPSI = −0.28). 

Higher coverage of exon 23 (E23), in the range of *GPR126*, was detected in IUGR samples, although the higher percentage of this exon usage (inclusion) was discovered in controls (Figure 5). Seven alternative first exon events with higher coverage ratios and PSI scores in controls libraries (-ΔPSI) were indicated in *TFPI*, *PSG6*, *CTRP1*, and *S100A13* (Table 4). Moreover, two promoter exons (AF) of TFPI were preferably used in IUGR-affected transcripts. Furthermore, the highest change in the percentage of transcripts with the exon spliced in (ΔPSI = 0.70) was calculated for *WARS*. Further, *PSG6* and *TFPI* harbored two additional AS events (AL and MXE) with differential splice junction usage (Table 4).

### 2.4. SNV Calling and RNA Editing Prediction

Based on reads aligned to the human genome, 88,859 SNVs candidates were discovered (Figure 6). After first filtering using standard GATK parameters, 72,275 SNVs were forwarded for further analysis. Substitutions located on bidirectional genes (24,877 SNVs), in the range of paralogous sequence (13,310), repetitive regions (2,529), close to any splice junction (723), and alternative allele fraction detected in a small number of biological replicates (23,184) were removed.

Among 7652 remaining allelic variants, 1,168 were significantly changed (FDR < 0.001) in alternative allele frequency (AAF) and were annotated according to chromosome localization, gene assignment, substitution type, variant functional effect, and transversion/transition ratio by SnpEff. Significant SNVs were overlapped with 6533 transcripts and assigned to 2386 CDS, 1528 introns, 345 5’UTR, 1260 3’UTR, 918 ncRNA, and 10 intergenic regions (Table 5). Within CDS regions, 543 missense substitutions with functional effects were recognized. The highest number of these changes occurred in downstream 5kb range out of 3’UTR (31.4%; Table 5). 

GO enrichment analysis of genes with nonsynonymous substitutions (FDR < 0.001) annotated *PIEZO1*, *PODXL*, *SWAP70* to “regulation of cell-cell adhesion mediated by integrin” biological process. Additionally, two known mutations (rs4575, synonymous; rs3817, on the downstream region) were detected within the range of AS and DEG candidates: *PA28β* and *PINLYP*, respectively. Among sites with significant AAF changes, four were qualified (frequency < 0.7 in all samples and no annotation in dbSNP, outside coding region) as the RNA editing candidates. RNA editing canonical candidates (adenosine to inosine, ‘A to I’) were localized within 3’ UTR of *EIF2AK2* and *CD99L2*, and within the intronic regions of *PPP1R13L* and *ARPC2* (Table 6).

## 3. Discussion

IUGR affects up to 15% of pregnancies [37]. Failure of deep placentation and maldevelopment of the placental villi, reduced cytotrophoblast proliferation, and capillarization constitute common pathologies of IUGR [38,39,40]. The proper placental vasculature is crucial for efficient transfer of nutrients, gases and waste elimination [40]. Imbalance of the immune system results in pathological pregnancies complicated by preeclampsia [41,42]. The condition is related to the increased inflammatory response that may contribute to disease progression [43,44]. IUGR is often complicated by preeclampsia that is manifested by multi-system disorder, as a result of severe maternal gestational hypertension with marked proteinuria [45,46]. Similar abnormalities for both diseases are associated with impaired interaction between the cytotrophoblast and the maternal spiral arteries, shallow placentation associated with endothelial dysfunction and impaired vascularization [47,48,49]. The functional insufficiency of the placenta may lead to miscarriage, fetal growth retardation or premature placental abruption [38]. Nearly one-third of IUGRs are due to genetic causes [4]. The aim of the current research was to present the global expression profile of human term placentas from pregnancies complicated by IUGR. We made an attempt to illustrate not only common differences in gene expression level but also deep transcriptomic changes in spliceosome and editome. Many known human diseases are frequently affected by alternative splicing events (ASEs) and genomic mutations [50]. Some disease-causing mutations (SNP) are conserved in human populations, others may appear spontaneously, i.e., during the transcriptional process. Especially RNA-editing sites may play a crucial role in the modulation of alternative isoforms splicing and consequently influence occurrence and course of diseases. Mutations might enhance or silence splice site signals of pre-mRNA, leading to the formation of the aberrant mRNA and protein products [51].

Pregnancy, complicated by IUGR, is considered to be a pro-inflammatory state with excessive activation of inflammatory cells and cytokines that may promote placental insufficiency [47]. Receptor for interleukin 7 (*IL7R*) codes the protein that plays a critical role during lymphocytes development [52,53]. Inhibiting of IL- 7/IL- 7R signaling pathway may contribute to maintaining a pregnancy [54]. Dysfunction of *IL7R* has been detected in patients with severe combined immunodeficiency (SCID) leading to chronic inflammatory diseases [55,56]. Revealed underexpression (log2FC = −2.07) of *IL7R* could have contributed to the immunological imbalance in IUGR-affected samples. Also, fibronectin type III domain containing 4 (*FNDC4*) dampen macrophage activation [57,58], and its robust upregulation results in the reduction of inflammation [59]. Considering mentioned functions, the identified underexpression of *FNDC4* (log2FC = −2.38), in the growth-restricted placentas, may be associated with progression of pro-inflammatory state contributing to the severity of the disease [57,59]. Gene expression analysis of placental microvascular endothelial cells indicates that butyrophilin like 9 (*BTNL9*) is significantly upregulated in IUGR [40]. *BTNL9* belongs to the butyrophilins (BTNs) family that comprises the molecules able to modulate immune homeostasis [60]. Our data confirmed the significant overexpression of *BTNL9* (log2FC = 2.43) in IUGR-affected placentas. It has been speculated that the upregulation of *BTNL9* in IUGR may oppose the inflammation associated with this pathology [40]. The IL2 inducible T cell kinase (*ITK*) regulates both the T and NK-cell development and signaling, influencing inflammatory cytokines secretion [61]. *ITK* seems to be indispensable in haematopoiesis as its deficiency cause immune dysregulation [62]. We observed decreased expression (log2FC = −2.03) of *ITK* in IUGR samples that is in agreement with the Zhong et al. [61], who indicated that deficiency of *ITK* leads to various disorders including malignancies, inflammation, and autoimmune diseases. *PINLYP* encodes phospholipase A2 inhibitor and LY6/PLAUR domain containing protein that may favor an anti-inflammatory effect [63]. Identified underexpression of *PINLYP* (log2FC = −1.8) may contribute to pro-inflammatory state. Moreover, we discovered rs3817 SNP (localized on the downstream region; < 5000 bp) close to *PINLYP* gene, indicating differential alternative allele frequency (ΔAAF), and such significant differences may be associated with *PINLYP* expression regulation. Additionally, we detected the underexpression (log2FC = −2.19) of beta-1,3-galactosyltransferase 5 (*B3GALT5*), that variants may be associated with fetal growth [64]. During pregnancy, inflammation may contribute to various disorders so further studies are needed to reveal how exactly the downregulation of mentioned gene affects the placental development leading to growth retardation of the fetus.

IUGR is a multifactorial disorder and any systemic infections might elicit changes in placental structure and might also be associated with increased risk for neurodevelopmental disorders [65,66]. B cell CLL/lymphoma 11B (*BCL11B*) is a transcriptional factor, required for the postnatal development of the hippocampus so the deficiency results in structural brain defects [67,68]. Patients with *BCL11B* mutations are affected by SCID and impaired T-cell development [68,69]. Underexpression of *BCL11B* (log2FC = −2.09) observed in studied IUGR-affected placentas may result in both immune and neuronal imbalance. PA28β proteasome activator subunit 2 (*PA28β*), a major regulator of the 20S proteasome [70], participates in protein degradation, regulation of cell cycle and antigen processing for the initiation of cell-mediated immunity [71]. Additionally, *PA28β* inhibits both cell growth and proliferation [72]. We detected ΔAAF in the synonymous site (rs4575) and occurrence of SE (skipping exon) within *PA28β* with decreasing value of PSI in IUGR samples. Mutations localized close to splice site signals may lead to the production of aberrant mRNA and finally altered protein product [50]. Further, signal regulatory protein gamma (*SIRPG*) is connected with systemic lupus erythematosus (SLE), one of the maternal risk factors of IUGR [6]. *SIRPG* expression during the active phase of SLE patients was significantly higher [73], however, our results indicated the downregulation (log2FC = −2.30) of *SIRPG*, so specific role in IUGR-affected placentas still needs to be explored. Additionally, we identified AAF change in *EIF2AK2*, the major marker of SLE [74]. Our stringent pipeline qualified this SNV as a candidate of RNA editing site associated with IUGR disorder. Another RNA editing candidates were localized within an intron of *PPP1R13L* (protein phosphatase 1 regulatory subunit 13 like) and *ARPC2* (actin related protein 2/3 complex subunit 2). *PPP1R13L* is significantly induced during *in vitro* trophectoderm differentiation [75], and *ARPC2* supports endothelial barrier during angiogenesis [76]. RNA editing site was also localized within 3’UTR of *CD99L2*, that is mostly expressed on leukocytes, endothelial cells and plays a role in extravasation of neutrophils under inflammatory conditions [77]. Selection and maturation of T-cells are regulated by both LCK proto-oncogene, Src family tyrosine kinase (*LCK*) and tyrosine kinases (*ZAP70*), as key signaling molecules essential for lymphokine production [78,79], moreover, *LCK* affects trophoblast invasion and decidualization [80,81]. Our analysis revealed underexpression of *ZAP70* (log2FC = −2.14) and *LCK* (log2FC = −2.13), what could have an impact on growth-retarded fetus. Deficient trophoblast invasion with the inadequate transformation of spiral arteries results in reduced uteroplacental blood flow and diminished placental oxygen levels [47,82]. Another gene found to be differentially-expressed in examined growth restricted placentas was the age-related maculopathy susceptibility 2 (*ARMS2*; log2FC = 3.30) that has been previously identified as highly expressed in the placenta [83,84]. Elevated levels of *ARMS2* protein detected in the growth-restricted twin placenta may be connected with dysfunction of the normal extracellular matrix [85]. Moreover, *ARMS2* might interact with pro-inflammatory molecules and impact on angiogenesis [83,86].

IUGR may be also induced through impaired placental nutrient transfer as highly associated with fetal malnutrition [87,88]. Solute carrier family 38 member 5 (*SLC38A5*) encodes SNAT5 protein, that mediates the cotransport of placental amino acids including glutamine, essential to cellular growth [89,90,91]. The altered expression of nutrient transporters is implicated in diverse mechanisms of fetal development [88]. Microarray analysis detected the *SLC38A5* expression as significantly upregulated however, the protein level was only slightly elevated in both IUGR and PE [88]. The current analysis detected downregulation of *SLC38A5* (log2FC = −2.28) in IUGR-affected placentas, that might be connected with the cellular metabolic status, modified in such heterogeneous disorder as IUGR [92]. Ligand-receptor interactions of TNF family members impact on placental cell death, growth, migration and hormone production [93]. Our analysis revealed downregulation (log2FC = −1.82) of lymphotoxin beta (*LTB*), a member of TNF family [94]. Pregnancy-specific glycoproteins (*PSGs*) are secreted by placental syncytiotrophoblasts [95] and contribute to the maintenance of successful pregnancy as low levels are associated with pathological complications including spontaneous abortion and IUGR [96]. *PSG6* have been recently found to be located within a genomic region enriched in segmental duplication of preeclampsia patients [97]. Within the range of *PSG6*, we observed alternative splicing of last (AL) and first (AF) exon, this may be associated with regulation of various isoforms expression and its consequences secretion of active proteins. SIN3B transcription regulator family member B (*SIN3B*) regulates the transcription of target genes to influence diverse cellular and biological functions such as embryonic development [98]. Results obtained in the present study provided ASE (skipping exon) within the region of *SIN3B* gene, and SE events revealed higher PSI values in IUGR samples. This may be related to the stability of protein isoforms and their divergent structures and functions [51]. Another transcriptomic marker of embryo development [99], ADAM metallopeptidase domain 2 (*ADAM2*) was overexpressed (log2FC = 2.56) within the IUGR-affected placenta. However, a significant decrease in *ADAM2* mRNA expression has been found in couples with the failure of in vitro fertilization, making it the valuable predictive marker of the efficiency of assisted reproductive technologies [99,100]. Another gene found to be upregulated in examined placentas was the age-related maculopathy susceptibility 2 (*ARMS2*; log2FC = 3.30). Elevated levels of ARMS2 protein detected in the growth-restricted twin placenta are suggested to be connected with dysfunction of the extracellular matrix [85]. Moreover, *ARMS2* interacts with pro-inflammatory molecules and impact on angiogenesis [101].

Analysis of SNP with recoding effect revealed *PIEZO1*, *PODXL*, and *SWAP70* as genes involved in the regulation of cell-cell adhesion mediated by integrins. Expression of integrins allows invasion of the trophoblasts into the endometrium and promotes placentation [102]. Several diseases, including IUGR, are caused by vascular problems that may be explained by anomalies in integrin patterns [103]. Together with the maternal uteroplacental circulation, the fetoplacental vasculature is essential for placental perfusion and determines fetal growth and successful pregnancy outcome [38,104]. Piezo type mechanosensitive ion channel component 1 (*PIEZO1*), is an important molecular marker of blood flow in the placenta and has the potential to be a new therapeutic target for the treatment of placental vascular diseases [105]. Switching B cell complex subunit SWAP70 is involved in regulating migration and invasion of trophoblast cells during the processes of embryonic implantation as well as placentation [106]. Podocalyxin (*PODXL*) is significantly increased in early-onset preeclampsia and may represent a novel marker of maternal endothelial cell dysfunction [107,108]. Taking under consideration such important functions of the above genes our results concerning recoding effect of identified SNPs should be further investigated.

Placental vascular pathologies represent multisystem disorders that may cause placental hypoperfusion, leading to IUGR [109], and affected placentas have significantly increased vascular lesions [110]. In the current analysis, we detected the set of angiogenesis-related genes associated with IUGR disorders. A pro-angiogenic factor (*S100A13*) is involved in angiogenesis by regulating cell signaling and endothelial cells metabolism [111]. Our investigation revealed decreased alternative usage of the first exon (AF) within *S100A13*. This alternative promoter change of *S100A13* may be essential in the adjustment processes of placental angiogenesis. Additionally, IUGR generated higher exon skipping rates within adhesion G protein-coupled receptor G6 (*GPR126*) gene, associated with endothelial cells of umbilical veins and angiogenesis [112]. Furthermore, a series of differential ASEs were identified on genomic loci of *TFPI* and *CTRP1*. Three splicing types (AF, SE, and MXE) detected within *TFPI* may alter the stability of the transcript causing unproductive placental splicing isoforms occurrence. Tissue factor pathway inhibitor (*TFPI*) regulates the extrinsic pathway of coagulation and plays a critical role in haemostasis [113]. Since *TFPI* is found in syncytiotrophoblasts, cytotrophoblasts, vascular endothelium, and macrophages of term pregnancy [114,115], the perturbations of placental expression may be involved in the occurrence of obstetrical vascular complications [116]. Furthermore, TFPI-deficient embryos die with signs of severe growth retardation [114]. We identified interesting spliceosome effect within *CTRP1* gene, affected by IUGR, revealed by underexpression of the alternative first exon, but with higher PSI value in the range of cassette exon. C1q/TNF-α–related protein 1 (*CTRP1*) exerts an anti-inflammatory effect by suppressing pro-inflammatory cytokine expression, and inflammatory responses in vascular cells [117,118]. CTRP1 functions as an adipokine to prevent the development of pathological vascular remodeling [119]. Disturbed regulation of alternative promoters, exon skipping, and mutually exclusive exons of these genes are valuable to elucidate IUGR etiology. Lower PSI value of SE event, detected within prosaposin (*PSAP*), may modulate gene function by removing protein domains and affecting protein activity. PSAP effects on the oligomerization of secreted PGRN monomers what might be detectable on protein-protein interaction [120]. Our data indicated that high *PGRN* expression, although not statistically significant, may be related to the splicing modulatory effect of *PSAP*. Trophoblastic expression of the *PGRN* was upregulated during the third trimester in cases of FGR and PE [121], however, the specific function of *PSAP* in placental vasculogenesis needs to be evaluated. Additionally, we detected the inclusion of sciellin (*SCEL*) cassette exon that may promote longer (active) protein isoform production. *SCEL* transcripts were upregulated by prolonged exposure to biomechanical stress in vascular smooth muscle cells [122]. Our investigation revealed that *SCEL* expression is regulated by alternative splicing, which may be affected by various IUGR factors. Tryptophanyl tRNA synthetase (*WARS*) has two variants that are produced via alternative splicing with anti-angiogenic activity [123]. WARS proteins were identified as downregulated in placentas complicated with gestational diabetes mellitus, one of the main risk factor of IUGR [6,124]. Our analysis enriched the present knowledge of *WARS*, indicating that increased exon usage of alternative promoter (AF) during pathological state suggest higher susceptibility of unknown transcription factor recruitment. Identified alternative variants should be further investigated to explore their potential role in IUGR pathobiology and the possibility to use as therapeutic targets.

## 4. Materials and Methods 

### 4.1. Ethics Statement

All clinical samples were collected at the Clinical Ward for Gynecology, Obstetrics, and Oncological Gynecology at the Regional Specialist Hospital in Olsztyn. The experimental protocol was approved by the Bioethics Committee of the Warmia-Mazury Medical Chamber (OIL. 164/15/Bioet) in Olsztyn, Poland. The hand-written informed consent, containing information about this study, was obtained from all of the pregnant women.

### 4.2. Clinical Characteristics of Placental Samples

The aim of the study was to compare the expression pattern of human term placenta between IUGR (*n* = 5) and control samples (*n* = 5). IUGR-affected placentas for the study group were collected from patients who underwent an elective or immediate cesarean section, due to the symptoms of the insufficiency of the fetal circulation or the abruption of the placenta. IUGR samples were confirmed not to bear any chromosomal abnormalities. Placentas for the control group were collected from healthy women, with no clinical abnormal signs of gestation, normal fetus growth, and development, at term pregnancy, who underwent a scheduled Cesarean section (CS) before the onset of labor due to states after previous CS. To reduce the sex bias, the placenta samples were derived from both fetus sexes. Placentas for the study group were collected from women with diagnosed IUGR. Fetal Growth Calculator (https://medicinafetalbarcelona.org/calc/) was applied to determine the percentiles based on estimated fetal weight, biparietal diameter, head circumference, abdominal circumference, and femur length that were appointed ultrasonographically and values of pulsatile index flows into medial cerebral artery, umbilical artery, maternal uterine arteries, resistance index, and systolic/diastolic ratio were calculated (GE Voluson 730). Abnormal fetal growth was recognized in these pregnancies where ultrasound EFW was below the 10th centile [8]. To diagnose IUGR, vessels flow should be estimated as a poor or cerebroplacental ratio below the 5th percentile or mean uterine artery pulsatility index higher than 95th percentile or correct flows but with EFW ≤ 3rd percentile, for given gestational age. For all patients, gestational age was confirmed by the assessing the first day of last menstrual period confirmed by the ultrasound scans with crown-rump length measurement performed during the first trimester of the pregnancy (between 8th–12th week) in 28-days cycles or only by ultrasound scans (between 8–10 weeks of pregnancy) in women with irregular cycles. Samples of placental tissues were collected immediately after delivery then, small pieces of placentas were frozen in liquid nitrogen. Preserved tissues were stored at −70 °C until RNA extraction.

### 4.3. Library Preparation and Sequencing Procedure

Total RNA was isolated from the placental tissues using the Qiagen RNeasy Kit according to the manufacturer’s recommendations. The RNase-Free DNase Set (Qiagen Venlo, The Netherlands) with DNA-ase digestion was used to obtain high-quality RNA. The total RNA concentration was calculated by Quant-IT RiboGreen (Invitrogen, Waltham, MA, USA). To assess the integrity of the total RNA, samples were run on the TapeStation RNA screentape (Agilent Technologies, Waldbronn, Germany). Only high-quality RNA preparations, with RIN greater than 8.0, were used for RNA library construction. A library was prepared with 1ug of total RNA for each sample by Illumina TruSeq mRNA Sample Prep kit (Illumina, Inc., San Diego, CA, USA). The first step in the workflow involved purifying the poly-A containing mRNA molecules using poly-T-attached magnetic beads. Following purification, the mRNA was fragmented into small pieces using divalent cations under elevated temperature. The cleaved RNA fragments were copied into first strand cDNA using SuperScript II reverse transcriptase (Invitrogen, Waltham, MA, USA) and random primers. This was followed by second strand cDNA synthesis using DNA Polymerase I and RNase H. These cDNA fragments then went through an end repair process, the addition of a single ‘A’ base, and then ligation of the indexing adapters. The products were then purified and enriched with PCR to create the final cDNA library. The libraries were quantified using qPCR according to the qPCR Quantification Protocol Guide (KAPA Library Quantification kits for Illumina Sequencing platforms) and qualified using the TapeStation D1000 ScreenTape (Agilent Technologies, Waldbronn, Germany). Indexed libraries were then sequenced using the HiSeq4000 platform (Illumina, San Diego, CA, USA). The sequencing data from this study have been submitted (https://www.ebi.ac.uk/ena) to the European Nucleotide Archive under accession no. PRJEB30656.

### 4.4. Quality Control and Mapping Processes

The raw paired-end reads quality was verified using the FastQC software v. 0.11.7 [www.bioinformatics.babraham.ac.uk]. Preprocessing using a Trimmomatic software v. 0.32 [125] included removal of Illumina adaptors and poly(A) stretches, exclusion of low-quality reads (Phred cutoff score ≤ 20; calculated on both ends of reads and with 10bp frameshift), and trimming of reads to equal length (90 bp). Next, paired-end clean reads were mapped to the reference human genome (Homo_sapiens.GRCh38.dna.primary_assembly.fa) with ENSEMBL/GENCODE annotation (Homo_sapiens.GRCh38.90.gtf) applying the STAR software v. 2.4 [https://github.com/alexdobin/STAR] mapper. Samtools software [126] was used for conversion BAM to SAM format. Further, Picard tool (with MarkDuplicates function) was used to remove multi-mapped reads and retain uniquely aligned reads in SAM file. StringTie v. 1.3.3 [https://ccb.jhu.edu/software/stringtie] [127] was used to prepare new annotations by merging Ensembl GTF file with reads mapped to the reference genome. Count expression values were estimated by ballgown software [128] and prepDE.py python script (stringtie module). All transcript sequences were extracted to FASTA file using a gffread script (https://github.com/gpertea/gffread). Mapping, SAM to BAM conversion and expression level calculation, as memory intensive processes, were performed in Regional IT Center of University of Warmia and Mazury in Olsztyn, applying 24-core CPU and 136 GB RAM server.

### 4.5. Transcriptome Profiling of Protein-coding Genes

Genome-wide transcriptome analysis of IUGR-affected placentas (Figure 1) was applied to identify transcriptome profiles and reveal the alternative regulation mechanism of spliceosome and editome. Count gene expression matrix was examined by two statistical methods, DESeq2 [129] and edgeR [130]. Both of them were applied to obtain the stringent results of differentially-expressed genes (DEGs). The changes in gene expression levels were considered as significant when statistical test values (adjusted *p*-value) were lower than 0.05. Candidate DEGs were visualized in an MA, volcano and heatmap plots with gplots [131] and ggplot2 [132] R Bioconductor packages [http://www.r-project.org/]. Obtained DEGs were scanned by Gene ontology (GO) and KEGG pathway databases using g.Profiler [133] software.

### 4.6. Alternative Splicing Events

We aimed to predict percent splicing inclusion (PSI) of the alternative exon, or changes of its inclusion (ΔPSI). Differential alternative splicing (DAS) events were explored by two compatible methods: a super-fast pipeline for alternative splicing analysis (SUPPA) [134] and the replicate multivariate analysis of transcript splicing (rMATS) [135]. Both software calculated ΔPSI score between IUGR-affected and control samples. Splicing analysis performed by SUPPA calculated PSI values directly from transcript isoform abundance. In order to receive SUPPA score, we extracted uniquely mapped reads from BAM files and remapped these reads to the reference transcriptome using Salmon software [136]. Next, SUPPA scripts generated potential splicing events and estimated ΔPSI values were statistically tested between biological replicates of IUGR-affected and controls samples. Standard parameters of rMATS v3.2.1 were applied to identify alternative splicing events (ASEs) using mapping to splice junctions and reads on target (ROT + JC) and splice junctions only (JC). Significant DAS events (FDR < 0.05) from rMATS and SUPPA were intersected using BEDTools software [137]. Obtained final set of DAS events were classified according to seven splicing types: alternative 5′splice site (A5SS), alternative 3′splice site (A3SS), alternative first exon (AF), alternative last exon (AL), mutually exclusive exons (MXE), retention intron (RI), and skipping exon (SE). Visualization of ASEs events was performed by maser package [138] and rmats2sashimiplot.py scripts [https://github.com/Xinglab/rmats2sashimiplot].

### 4.7. SNV calling and RNA Editing Sites Prediction

Picard tool [http://picard.sourceforge.net] was applied in the recalibration process to filter out reads that may constitute a noise disturbing single nucleotide variants (SNVs) calling. Variant calling analysis was performed by rMATS-DVR [139] and GATK tool (the golden standard for variant detection by NGS) [140]. Heterozygous genomic position of each potential SNV was statistically proven by UnifiedGenotyper and collected in Variant Call Format (VCF). Identified SNVs were described, for IUGR-affected and control samples, by allele and reference fraction, allele fraction difference, *p*-value, FDR value, chromosomal position, gene, and single nucleotide polymorphism (SNP) annotation. To predict RNA editing site, R Bioconductor multi-step analysis was applied to filter previously detected SNVs. First, SNVs were filtered out according to standard GATK parameters: total depth of base coverage <10; RMSMappingQuality <40; QualitybyDepth < 2; MappingQualityRankSum < −12.5; and ReadPosRankSum < −8. Moreover, all SNVs located within: 5bp intronic flanking region, bidirectional transcription regions (sense and antisense genes), and simple sequence repeat (SSR) regions (with +/− 3 bp range) identified by GMATo tool [141] were filtered out. SNVs, along with 50 bp upstream and downstream sequences, were aligned to the human genome by BLAT [142], to eliminate multi-aligned SNVs, localized within repetitive regions or paralogs. Further, SNVs displaying alternative allele fraction = 0, in at least five libraries, were also excluded from the dataset. According to the assumption of the next filter, the occurrence of alternative allele must have been observed in more than half samples of the experiment. To point out variants related to IUGR disorder only SNVs with significant allele differentiation (FDR < 0.001) were selected to the annotation step. Survived SNVs were tested by VEP [143] and SnpEff [144] to predict the effects of variants. Annotated variants were divided into potential RNA editing sites and SNVs associated with IUGR. We qualified variants as potential RNA editing site when the frequency of alternative allele was < 0.7 in all samples, the substitution was canonical (A-to-I and C-to-U) and the site was unclassified in dbSNP. The second pool consisted of known variants (rs_ID in dbSNP), noncanonical substitutions, intergenic sites and all variants with alternative allele frequency > 0.7 at least one sample. Circos software package [145] was implemented to visualize all IUGR-affected variants. Additionally, to identify the functions of the genes harboring SNVs, GO analysis was performed.

## 5. Conclusions

Despite progress in the field of molecular diagnosis, there is still an urgent need to develop detection, prevention, and management of IUGR-affected pregnancies to improve their outcomes. Investigating the molecular mechanisms underlying the IUGR could be useful for better understanding of placental signaling pathways. This severe gestational condition still lacks direct causation and is a valid scientific target. We pointed novel genes that are dysregulated in IUGR, which can serve as a basis for identifying the mechanisms underlying pathological pregnancies, as well as may be useful for early detection of the genomic defects. The tremendous efforts should be undertaken to decipher their implication in the course of gestation, placenta development and reproductive disorders.

## Figures and Tables

**Figure 1 ijms-20-01510-f001:**
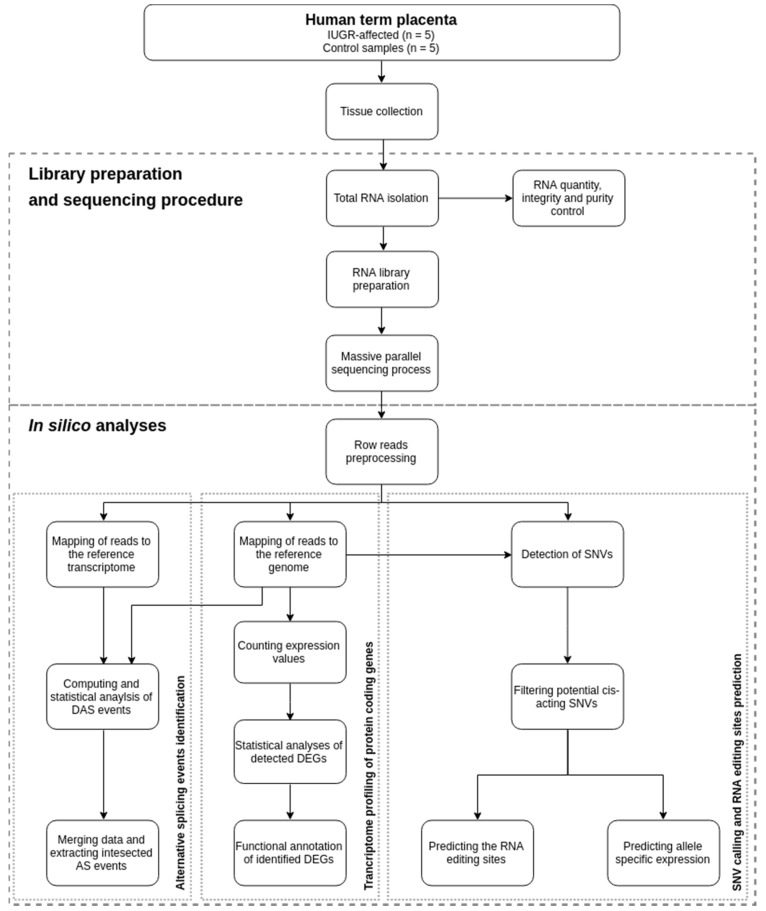
Flowchart of sequencing, computational analysis, and validation procedures.

**Figure 2 ijms-20-01510-f002:**
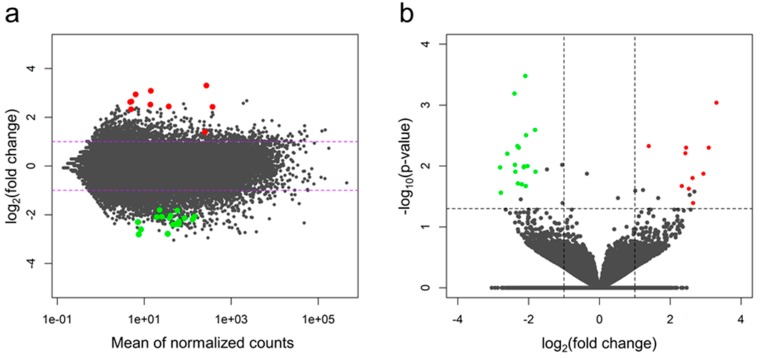
(**A**) MA plot presenting logarithmic values of fold change plotted against normalized counts for IUGR-affected and control libraries. Dotted horizontal lines indicate applied fold change (absolute value of log2FC >1) cut-off. (**B**) Volcano plot showing log2FC plotted against normalized *p*-values. Dotted horizontal line indicates negative logarithmic adjusted *p*-value (0.05) cut-off. Dotted vertical lines indicate applied fold change (absolute value of log2FC >1) cut-off. Each point (dot) represents gene expression value; green (underexpressed in IUGR) and red (overexpressed in IUGR) represent the significant DEGs (adjusted *p*-value < 0.05). Both plots present values (points positions) calculated with the DESeq2 algorithm, green and red points depict DEGs confirmed by both DESEq2 and edgeR methods. Grey dots indicate nonsignificant genes.

**Figure 3 ijms-20-01510-f003:**
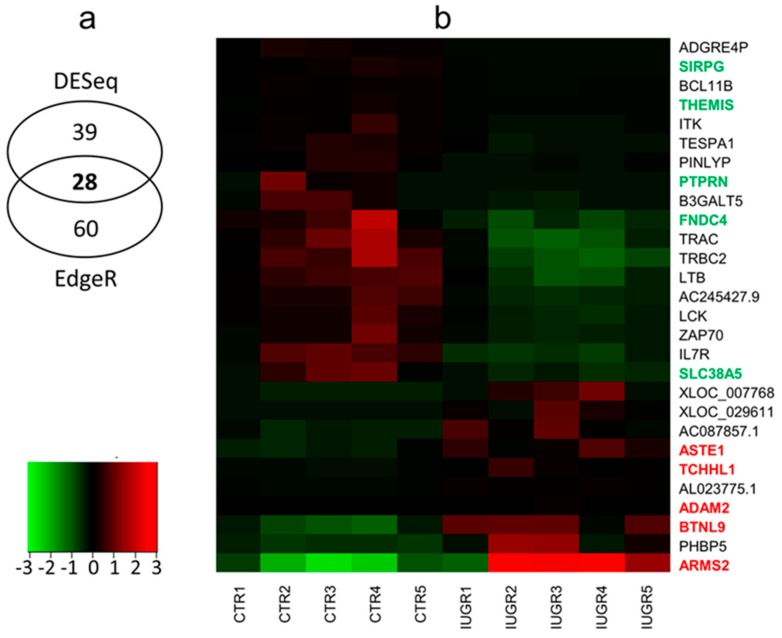
(**A**) Venn diagram with the number of differentially-expressed genes (DEGs) identified by DESeq2 and edgeR. Middle number indicates common DEGs confirmed by both methods. (**B**) Heatmap of expression data for the 28 DEGs (for definitions of gene symbols, see Table 2). Columns represent individual libraries; rows indicate gene symbols of DEGs. The z-score scale was applied for visualization expression values (FPKM) of each biological replicate.

**Figure 4 ijms-20-01510-f004:**
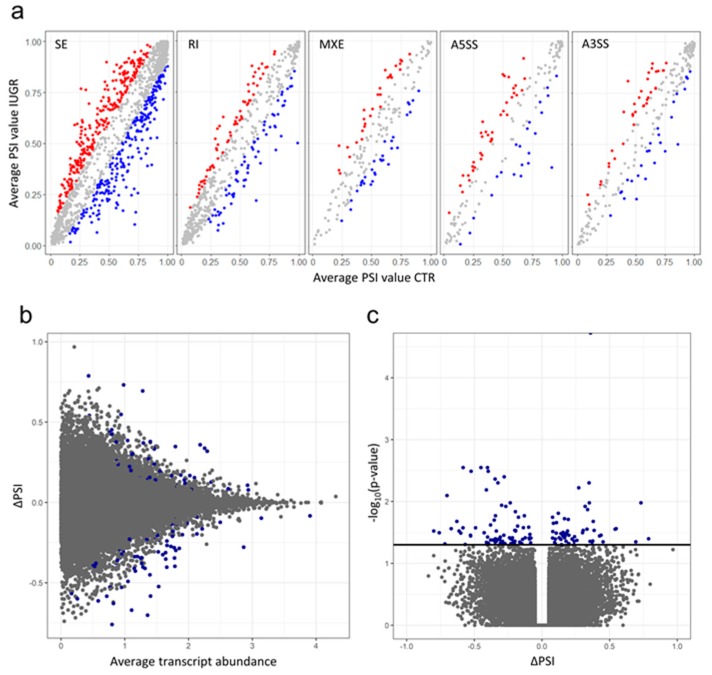
The abundance of alternative splicing events. (**A**) Dot plots for five alternative splicing (AS) events types depicting imbalance in percent splicing inclusion (PSI) values investigated by rMATs. Red and blue dots indicate significant AS events with higher PSI value in IUGR-affected and control libraries, respectively. Grey dots represent no changes in PSI ratio. (**B**) MA plot showing changes in the percentage of transcripts with the exon spliced in (ΔPSI) plotted against average transcript abundance for all AS events detected in IUGR-affected and control libraries retrieved by SUPPA. (**C**) ΔPSI values of each AS event were presented on X-axis and the negative logarithmic adjusted *p*-value was presented on the Y-axis. Horizontal line indicates negative logarithmic adjusted *p*-value (0.05) cut-off. Each point (dot) on Volcano and MA plot represents the splicing event; blue dots represent significant DAS events (adjusted *p*-value < 0.05) and grey dots were not significant.

**Figure 5 ijms-20-01510-f005:**
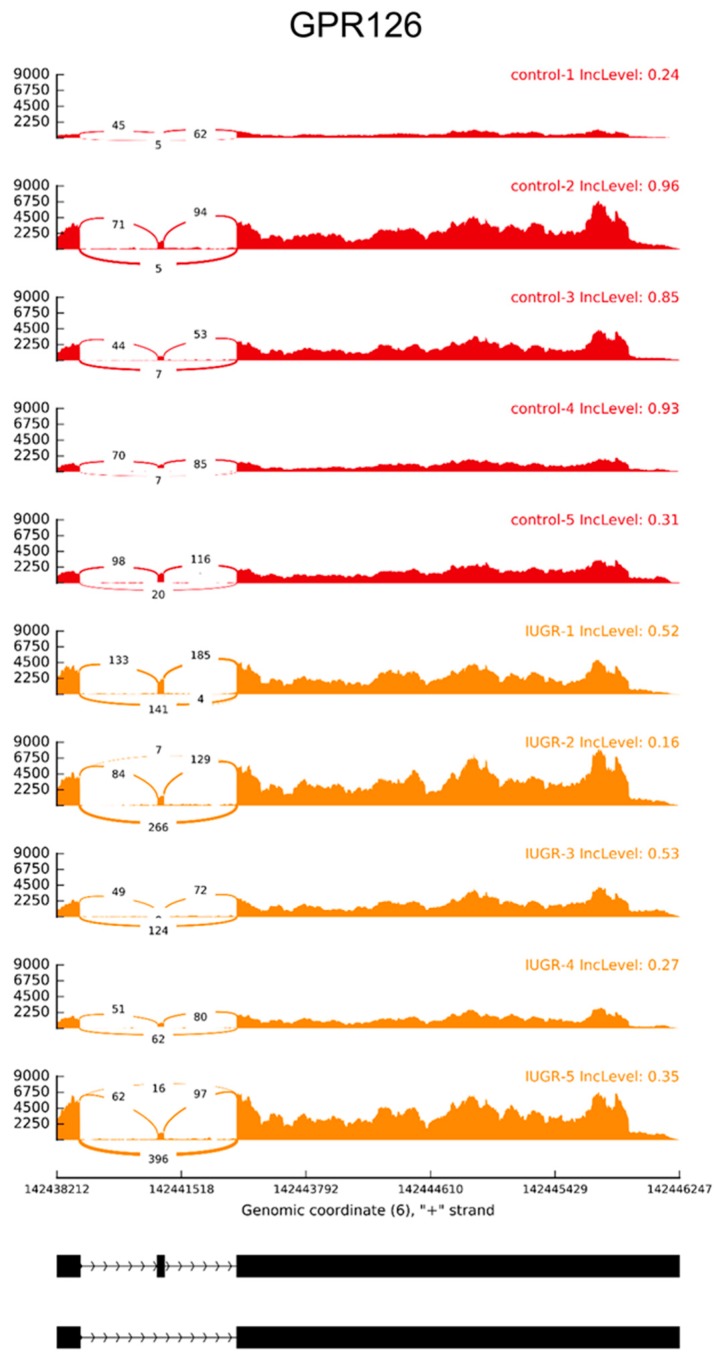
Quantitative visualization (Sashimi plot) of an alternative skipping event of exon 23 (E23) within the *GPR126* (*ADGRG6*). Ten upper tracks represent biological replicates for control (red) and IUGR (orange) libraries. Numbers on curved lines indicate counts engaged in each splice junction. The scale on left present coverage depth of GPR126 in range of ASE region. Percent splicing inclusion (PSI) present on the right side of each sample in the upper track. Middle track depicts genomic coordinates of ASE on chromosome 6. The bottom track represents exon-intron structures of DAS event in the range of E23.

**Figure 6 ijms-20-01510-f006:**
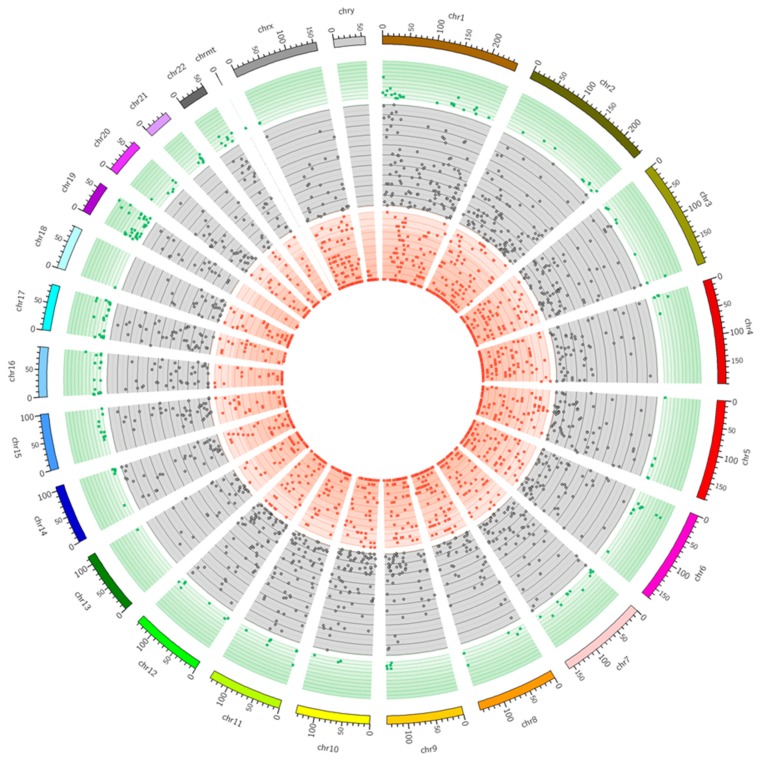
Distribution of SNVs detected on the physical map of the human genome designated by rMATS-DFR; scale in Mb. The red circle indicates the regions with small numbers (< 15) of SNVs per 1 Mb. The gray circle indicates the regions with medium numbers (>15 and <100) of SNVs. The green circle indicates the richest regions of SNVs (>100).

**Table 1 ijms-20-01510-t001:** Sequencing, quality control and mapping metrics for each placental control (CTR) and IUGR-affected library. Mapped reads were divided to uniquely (reads that map exactly one time on the genome), multi-mapped (reads that map 2–20 times on the genome) and to too many loci (reads that map more 20 times on the genome).

	CTR1	CTR2	CTR3	CTR4	CTR5	IUGR1	IUGR2	IUGR3	IUGR4	IUGR5
**Fetal sex**	female	female	female	male	male	female	female	male	male	male
**Row reads**	64,041,868	76,079,178	55,220,598	73,481,438	57,398,092	43,095,034	56,174,786	36,854,904	42,138,704	41,532,984
**Trimmed reads**	57,087,670	66,388,446	47,395,044	64,759,848	49,765,752	37,894,430	49,348,796	30,969,750	36,987,266	36,422,794
**Mapped**	54,927,730	64,035,634	45,653,626	62,541,870	48,246,002	36,523,434	47,657,008	29,887,328	35,801,520	35,037,148
**Uniquely mapped**	50,834,558	53,206,552	41,489,480	60,190,958	44,263,500	33,625,168	42,868,410	28,014,300	32,690,420	32,541,876
**Multi-mapped**	4,082,300	10,817,500	4,156,886	2,342,936	3,974,602	2,891,872	4,780,564	1,868,814	3,104,808	2,488,174
**Too many loci**	10,872	11,582	7260	7976	7900	6394	8034	4214	6292	7098

**Table 2 ijms-20-01510-t002:** List of 28 differentially-expressed genes (under and over-expressed) between IUGR-affected and control libraries.

Gene_ID	BaseMean	log2FC	Padj	Gene Symbol
XLOC_048584	7.49193	−2.805	1.05 × 10^−2^	THEMIS
XLOC_032992	34.69158	−2.784	2.74 × 10^−2^	PTPRN
XLOC_028055	8.40907	−2.603	6.26 × 10^-3^	ADGRE4P
XLOC_050480	45.34250	−2.401	6.47 × 10^−4^	AC245427.9
XLOC_031488	63.22940	−2.386	9.55 × 10^-3^	FNDC4
XLOC_015723	52.46403	−2.375	1.24 × 10^−2^	TRAC
XLOC_050482	62.93612	−2.313	4.71 × 10^-3^	TRBC2
XLOC_034173	7.15533	−2.304	1.93 × 10^−2^	SIRPG
XLOC_058872	62.88279	−2.277	5.00 × 10^−3^	SLC38A5
XLOC_035229	129.54309	−2.191	1.99 × 10^−2^	B3GALT5
XLOC_030101	84.31897	−2.137	1.05 × 10^−2^	ZAP70
XLOC_000516	38.22445	−2.130	1.02 × 10^−2^	LCK
XLOC_017361	19.31762	−2.093	3.33 × 10^−4^	BCL11B
XLOC_013236	25.54894	−2.078	2.13 × 10^−2^	TESPA1
XLOC_043316	143.34412	−2.072	3.12 × 10^−3^	IL7R
XLOC_044222	40.17459	−2.033	1.00 × 10^−2^	ITK
XLOC_047901	57.73795	−1.817	2.56 × 10^−3^	LTB
XLOC_027402	22.55203	−1.804	1.24 × 10^−2^	PINLYP
XLOC_039524	248.13131	1.392	4.71 × 10^−3^	ASTE1
XLOC_048872	4.95620	2.326	2.13 × 10^−2^	AL023775.1
XLOC_044458	371.82484	2.429	6.17 × 10^−3^	BTNL9
XLOC_037183	36.60827	2.447	5.00 × 10^−3^	AC087857.1
XLOC_004057	13.87959	2.524	2.36 × 10^−2^	TCHHL1
XLOC_029611	4.74119	2.629	1.57 × 10^−2^	NA
XLOC_053736	5.04141	2.647	4.06 × 10^−2^	ADAM2
XLOC_007768	6.30889	2.937	1.34 × 10^−2^	NA
XLOC_051256	14.10328	3.087	5.00 × 10^−3^	PHBP5
XLOC_006484	268.76888	3.304	9.15 × 10^−4^	ARMS2

**Table 3 ijms-20-01510-t003:** Gene Ontology (GO) and Kyoto Encyclopedia of Genes and Genomes (KEGG) analysis of differentially-expressed genes identified in IUGR-affected placentas.

Term ID	Term Name	Gene Symbol
GO:0050851	antigen receptor-mediated signaling pathway	ITK, ZAP70, TESPA1, THEMIS, LCK
GO:0050852	T cell receptor signaling pathway	ITK, ZAP70, TESPA1, THEMIS, LCK
GO:0007159	leukocyte cell-cell adhesion	SIRPG, ZAP70, TESPA1, IL7R, LCK
GO:1903037	regulation of leukocyte cell-cell adhesion	SIRPG, ZAP70, TESPA1, IL7R, LCK
GO:0045321	leukocyte activation	SIRPG, ITK, ZAP70, BCL11B, TESPA1, IL7R, THEMIS, LCK
GO:0046649	lymphocyte activation	SIRPG, ITK, ZAP70, BCL11B, TESPA1, IL7R, THEMIS, LCK
GO:0042110	T cell activation	SIRPG, ITK, ZAP70, BCL11B, TESPA1, IL7R, THEMIS, LCK
GO:0002520	immune system development	ITK, ZAP70, BCL11B, TESPA1, IL7R, THEMIS, LCK, LTB
GO:0048534	hematopoietic or lymphoid organ development	ITK, ZAP70, BCL11B, TESPA1, IL7R, THEMIS, LCK, LTB
GO:0030097	hemopoiesis	ITK, ZAP70, BCL11B, TESPA1, IL7R, THEMIS, LCK,
GO:0022409	positive regulation of cell-cell adhesion	SIRPG, ZAP70, TESPA1, IL7R, LCK
GO:1903039	positive regulation of leukocyte cell-cell adhesion	SIRPG, ZAP70, TESPA1, IL7R, LCK
GO:0050863	regulation of T cell activation	SIRPG, ZAP70, TESPA1, IL7R, LCK
GO:0002521	leukocyte differentiation	ITK, ZAP70, BCL11B, TESPA1, IL7R, THEMIS, LCK
GO:0030098	lymphocyte differentiation	ITK, ZAP70, BCL11B, TESPA1, IL7R, THEMIS, LCK
GO:0030217	T cell differentiation	ITK, ZAP70, BCL11B, TESPA1, IL7R, THEMIS, LCK
GO:0045058	T cell selection	ZAP70, BCL11B, THEMIS
GO:0043368	positive T cell selection	ZAP70, BCL11B, THEMIS
GO:0033077	T cell differentiation in thymus	ZAP70, BCL11B, TESPA1, IL7R
GO:0051251	positive regulation of lymphocyte activation	SIRPG, ZAP70, TESPA1, IL7R, LCK
GO:0050870	positive regulation of T cell activation	SIRPG, ZAP70, TESPA1, IL7R, LCK
GO:0004715	non-membrane spanning protein tyrosine kinase activity	ITK, ZAP70, LCK
KEGG:04064	NF-kappa B signaling pathway	ZAP70, LCK, LTB
KEGG:04660	T cell receptor signaling pathway	ITK, ZAP70, LCK
KEGG:05340	Primary immunodeficiency	ZAP70, IL7R, LCK

**Table 4 ijms-20-01510-t004:** Statistically significant AS events identified by Suppa (*p*-value < 0.05) and intersected with rMATs (adjusted *p*-value < 0.05).

Gene Symbol	AS Type	ΔPSI	*p*-value	AS Genomic Coordinates
TFPI	AF	0.35943	0	2:187503770-187513616:187513641:187503770-187554200:187554492:-
TFPI	AF	0.33763	1.349 × 10^−2^	2:187503770-187513616:187513653:187503770-187554200:187554492:-
TFPI	AF	−0.23995	3.853 × 10^−2^	2:187503770-187513616:187519860:187503770-187554200:187554438:-
WARS	AF	0.69358	4.496 × 10^−2^	14:100369258-100375283:100375473:100369258-100376260:100376308:-
PSG6	AF	−0.20069	2.922 × 10^−2^	19:42910858-42916125:42916398:42910858-42917729:42917887:-
C1QTNF1	AF	−0.31195	3.996 × 10^−2^	17:79022934:79023088-79043955:79025663:79025935-79043955:+
C1QTNF1	AF	−0.21092	4.396 × 10^−2^	17:79024070:79024494-79043955:79025663:79025935-79043955:+
C1QTNF1	AF	−0.33645	4.396 × 10^−2^	17:79024261:79024494-79046555:79043593:79044123-79046555:+
S100A13	AF	−0.22921	4.795 × 10^−2^	1:153626533-153626951:153627048:153626533-153628121:153628180:-
S100A13	AF	−0.20534	4.795 × 10^−2^	1:153626533-153627139:153627268:153626533-153628121:153628180:-
PSG6	AL	−0.20069	2.922 × 10^−2^	19:42906636:42907176-42910580:42907725:42907854-42910580:-
TFPI	MX	−0.21823	3.853 × 10^−2^	2:187503770-187513616:187513646-187554200:187503770-187520542:187520613-187554200:-
TFPI	SE	0.12683	3.853 × 10^−2^	2:187503770-187529364:187529485-187554200:-
EVI5	SE	−0.30817	4.795 × 10^−2^	1:92636336-92663420:92663452-92665939:-
PSME2	SE	−0.28553	4.895 × 10^−2^	14:24145772-24146208:24146240-24146534:-
ADGRG6	SE	−0.28155	3.497 × 10^−2^	6:142438364-142440908:142440953-142443337:+
SIN3B	SE	0.09338	4.595 × 10^−2^	19:16862559-16862884:16862979-16863680:+
SCEL	SE	0.35527	4.412 × 10^−2^	13:77591460-77593514:77593573-77597545:+
C1QTNF1	SE	0.23831	4.396 × 10^−2^	17:79024494-79043955:79044123-79046555:+
PSAP	SE	−0.09818	2.597 × 10^−2^	10:71822007-71823888:71823896-71825837:-

**Table 5 ijms-20-01510-t005:** Distribution (%) of significant SNV (ΔAFF; FDR < 0.05) annotated according to gene components (upstream region, 5’UTR, CDS, intron, 3’UTR, and downstream region).

SNV Localization	Number of SNV Annotations	Percentage Ratio
DOWNSTREAM	3179	31.38%
EXON	2386	23.55%
INTERGENIC	10	0.10%
INTRON	1528	15.08%
SPLICE_SITE_ACCEPTOR	1	0.01%
SPLICE_SITE_REGION	54	0.53%
UPSTREAM	1369	13.51%
UTR_3_PRIME	1260	12.44%
UTR_5_PRIME	345	3.41%

**Table 6 ijms-20-01510-t006:** Four main RNA editing candidates identified in IUGR-affected samples.

Chr	Site	Quality	FDR	Fraction Difference	Gene Name	Strand	Location	Subst.
19	45394416	357.14	9.99 × 10^−5^	0.122	PPP1R13L	-	Intron	A>G
2	37100515	373.16	1.06 × 10^−5^	0.106	EIF2AK2	-	3UTR	A>G
2	218237842	275.35	7.53 × 10^−4^	0.178	ARPC2	+	Intron	A>G
X	150767753	438.58	2.39 × 10^−4^	0.047	CD99L2	-	3UTR	A>G

## Data Availability

The sequencing data from this study have been submitted (https://www.ebi.ac.uk/ena) to the European Nucleotide Archive under accession no. PRJEB30656.

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
