# Peer review of "Placenta Transcriptome Profiling in Intrauterine Growth Restriction (IUGR)"

_ijms, 2019, doi:10.3390/ijms20061510_

Round 1

Reviewer 1 Report

The authors have presented RNA seq analysis of IUGR placenta compared to normal control placenta and have explained the genetic background to IUGR complications. the article is presented well and the authors need to address some minor issues listed below

Expand CTR in table 1 legend

In figure 2 legend explain red and green (overexpressed and underexpressed respectively) in which condition (control or IUGR)

In figure 3B highlight the names of the genes mentioned in the text (five up and five down regulated)

In the gene ontology analysis shown in table 3, please explain if there is a pattern or functional differentiation of the 10 DEG (up and down) clearly shown to be different between Control and IUGR in Fig 3B

In line 214 expand DAS 

In line 215 expand ASE or change in line 212

If the methods sections are to be placed in the end following the results and discussion sections, the abbreviations, SE, RI, MXE, A5SS and A3SS needs to be expanded in the text in line 217. 

Author Response

The authors have presented RNA seq analysis of IUGR placenta compared to normal control placenta and have explained the genetic background to IUGR complications. the article is presented well and the authors need to address some minor issues listed below

Authors would like to thank the Reviewer for the positive opinion regarding our work. We are very grateful for all the suggestions and comments that were more than valuable and allowed us to improve the manuscript. Below, please, find our reply to the Reviewer comments.

1. Expand CTR in table 1 legend

According to the Reviewer suggestion, the CTR abbreviation was expanded in Table 1 legend (lines 147 – 148).

2. In figure 2 legend explain red and green (overexpressed and underexpressed respectively) in which condition (control or IUGR)

According to the Reviewer comment, the proper explanation was included in Figure 2 legend (lines 166 – 167).

3. In figure 3B highlight the names of the genes mentioned in the text (five up and five down regulated)

Mentioned up and down-regulated genes were marked in red and green, respectively in Figure 3B.

4. In the gene ontology analysis shown in table 3, please explain if there is a pattern or functional differentiation of the 10 DEG (up and down) clearly shown to be different between Control and IUGR in Fig 3B

Table 3 described biological processes that fulfilled cut-off (0.05) enrichment parameter in GO analysis, and only downregulated DEGs created biological networks that were qualified as significant. Downregulated DEGs were engaged mainly in T cell regulation and in the immune response. Adequate information was included in the description of the results (lines 212 – 214).

5. In line 214 expand DAS

According to the Reviewer recommendation, DAS was explained (line 238). Also, all abbreviations were explained as early as the first mention.

6. In line 215 expand ASE or change in line 212

Adequate changes were introduced (lines 236).

7. If the methods sections are to be placed in the end following the results and discussion sections, the abbreviations, SE, RI, MXE, A5SS and A3SS needs to be expanded in the text in line 217.

Due to Reviewer remark, all abbreviations were expanded when they were used for the first time.

Reviewer 2 Report

This manuscript provides transcriptome data in placenta with IUGR to help characterize molecular pathway involved in IUGR. The authors identify 28 genes expressed differentially in placenta with IUGR. Although the biological importance of altered expression level of gene(s) should be verified in further experiments, the manuscript contains interesting findings for studying the molecular mechanism underlying IUGR. 

There are minor concerns

(1) Did women with IUGR also undergo Cesarean Section before the onset of labor? 

(2) Fetal sex should be described in all samples.

(3) The discussion is excessively long and should be shortened.

Author Response

Authors wish to thank the Reviewer very much for reading the manuscript and providing valuable comments. Below, please, find our reply to the Reviewer comments.

There are minor concerns

1. Did women with IUGR also undergo Cesarean Section before the onset of labor?

IUGR-affected placentas for the study group were collected from patients who underwent an elective or immediate cesarean section, due to the symptoms of the insufficiency of the fetal circulation or the abruption of the placenta. This missing information was included in the Materials and Methods section (lines 586 – 588).

2. Fetal sex should be described in all samples.

Information regarding fetal sex was provided in Table 1.

3. The discussion is excessively long and should be shortened.

We have to admit that discussion is extensive. Due to the Reviewer suggestion, we have made every attempt to shorten this section, but we came to the conclusion that it is a reflection of the number of obtained results, and appropriate, valid discussion is needed. We respect the Reviewer opinion in this manner, but we were not able to shorten the Discussion.

Reviewer 3 Report

The manuscript by Majewska and colleagues represents a continuation of their previous work on the expression of placental genes. They had previously described the transcriptome profile of human placenta and analyzed long non-coding RNA expression in human placenta. In this manuscript, the authors have analyzed the difference in expression profiles of placentas in intra-uterine growth restriction cases. This severe condition still lack direct causation and is a valid scientific target.

I consider that the experiment and the analysis are well performed and that the conclusions made are valid. On the other hand, I feel that the main message could be pin-pointed in a better and more interesting manner. Thus I would suggest improving the abstract ending and conclusion part.

Besides minor comments below, I suggest the article be published as is.

Minor comments:

1.       I would suggest rewriting the ending of abstract to clearly send the main message of the study and the results.

2.       Line 129: The percentages do not add up to 100%, intronic and intergenic numbers are not as in Fig S1.

3.       Figure 2 is lacking info about which algorithm was used for plotting (DESeq or edgeR). Also Methods section states DESeq2, main text lacks number 2.

4.       I would suggest that genes in Table 2 are sorted by expression level and not by Gene_ID

5.       Frequently some abbreviations do not have explanation until for example Methods section, and explanations are not included in figure legends (e.g. A5SS which is in Fig 4 at line 225, but explanation “5’ alternative splicing site” is included in Methods at line 634). Please put explanation as early as first mention of the abbreviation. Some abbreviations do not have explanations at all (e.g. CDS).

6.       Statement in line 264 seems contrary to inclusion levels in Fig 5. The authors should either correct it or write a clearer statement that corresponds to data presented. Also, line 306 in the Fig 5 legend states “Two upper tracks…” when instead there are 10 tracks that this statement refers to.

7.       Line 313 needs “with” to be removed

8.       “Among 7,652 survived allelic variants” would sound better as “Among 7,652 remaining allelic variants”

9.       The authors clearly state that IUGR is often a result of chromosomal abnormalities but have not stated anywhere if the etiology of observed IUGRs used in the study was checked for it. I believe that this is an important information and should be clearly stated in Methods section whether IUGR samples were confirmed not to bear any chromosomal abnormalities.

Author Response

Authors are very grateful for Reviewer suggestions and comments that were more than valuable and allowed us to improve our manuscript. Below, please, find our reply to the Reviewer comments.

Besides minor comments below, I suggest the article be published as is.

Minor comments:

1. I would suggest rewriting the ending of abstract to clearly send the main message of the study and the results.

Authors made every effort to improve the manuscript due to the Reviewer’s requests. Appropriate changes were done at the end of the abstract (lines 38 – 40) and conclusion part (lines 711 – 719).

2. Line 129: The percentages do not add up to 100%, intronic and intergenic numbers are not as in Fig S1.

Authors are very grateful for indicating this oversight, the proper numbers were included in the text according to Fig. S1 data, and now the percentages add up are equal to 100% (lines 134 – 135).

3. Figure 2 is lacking info about which algorithm was used for plotting (DESeq or edgeR). Also Methods section states DESeq2, main text lacks number 2.

Authors included lacking information about the algorithm used for plotting (lines 167 – 169), also “2” was added in lacking positions (lines 153, 190).

4. I would suggest that genes in Table 2 are sorted by expression level and not by Gene_ID

According to the Reviewer's suggestion, genes in Table 2 were sorted by expression level.

5. Frequently some abbreviations do not have explanation until for example Methods section, and explanations are not included in figure legends (e.g. A5SS which is in Fig 4 at line 225, but explanation “5’ alternative splicing site” is included in Methods at line 634). Please put explanation as early as first mention of the abbreviation. Some abbreviations do not have explanations at all (e.g. CDS).

Authors would like to apologize for this neglect. All abbreviations were explained as early as the first mention.

6. Statement in line 264 seems contrary to inclusion levels in Fig 5. The authors should either correct it or write a clearer statement that corresponds to data presented. Also, line 306 in the Fig 5 legend states “Two upper tracks…” when instead there are 10 tracks that this statement refers to.

Authors are very grateful for this remark. Appropriate Fig. 5 was provided and now all data are in agreement. The mistake was done during sashimi plot generation. All information included in the manuscript were described according to alternative splicing data (tabular output files), obtained from Suppa and rMATS tools. Both software confirmed negative ΔPSI values within exon 23 of the GPR126 in the comparison between IUGR and control samples. The PSI values for this event were compatible with both methods. The appropriate changes were also made in Figure 5 legend (lines 337 – 338).

7. Line 313 needs “with” to be removed

The indicated change was introduced (line 349).

8. “Among 7,652 survived allelic variants” would sound better as “Among 7,652 remaining allelic variants”

The text was corrected accordingly to the Reviewer comment (line 378).

9. The authors clearly state that IUGR is often a result of chromosomal abnormalities but have not stated anywhere if the etiology of observed IUGRs used in the study was checked for it. I believe that this is an important information and should be clearly stated in Methods section whether IUGR samples were confirmed not to bear any chromosomal abnormalities.

No chromosomal abnormalities were diagnosed in analyzed samples. Appropriate information was supplied in the Materials and Methods section (lines 588 – 589).
